# Environmentally Responsible Purchase Intention in Pacific Alliance Countries: Geographic and Gender Evidence in the Context of the COVID-19 Pandemic

**DOI:** 10.3390/bs13030221

**Published:** 2023-03-03

**Authors:** Elizabeth Emperatriz García-Salirrosas, Juan Carlos Niño-de-Guzmán, Ledy Gómez-Bayona, Manuel Escobar-Farfán

**Affiliations:** 1Faculty of Management Science, Universidad Autónoma del Perú, Lima 15842, Peru; 2Faculty of Business and Legal Sciences, Universidad de Montemorelos, Montemorelos 67515, Nuevo Leon, Mexico; 3Faculties Business, Universidad de San Buenaventura, Medellín 050010, Colombia; 4Department of Administration, Faculty of Administration and Economics, University of Santiago of Chile, Santiago 9170020, Chile

**Keywords:** environmental awareness, sustainable consumption, social responsibility, Pacific Alliance

## Abstract

The objectives of this research were: (1) to examine the influence of environmental awareness (EA), sustainable consumption (SC) and social responsibility (SR) on the environmentally responsible purchase intention (ERPI) of consumers in the member countries of the Pacific Alliance, namely, Chile, Colombia, Mexico and Peru; and (2) to analyze whether there is a moderating effect related to the country of residence and gender of the consumer. The study was conducted under a quantitative and cross-sectional approach. The sample consisted of 1646 consumers: 24.4% from Peru (n = 402), 25.4% from Mexico (n = 418), 26.1% from Colombia (n = 401) and 24.1% from Chile (n = 397). Data analysis and hypothesis testing were performed using a multigroup Structural Equation Model (SEM). The results show a positive influence among environmental awareness (EA), sustainable consumption (SC) and social responsibility (SR) on environmentally responsible purchase intention (ERPI). Gender and country of residence were also shown to be moderating variables in these relationships. In conclusion, it can be affirmed that the participants of this study recognize the importance of acquiring environmentally friendly products. Among them, the female population is more aware of this issue. It is recommended new business models be created to provide products and services oriented to this market according to consumers’ tastes, desires and purchasing preferences; the proposals they have should be friendly to the environment and to society.

## 1. Introduction

The restrictions due to the COVID-19 pandemic have generated a tremendous economic and social impact, altering lifestyles, creating environmental awareness, modifying social actions and promoting new trends in global consumption [1,2,3]. In order to reduce contagion, distancing measures and avoiding real contact between people have driven the exponential growth of electronic commerce, accelerating an undoubted transition toward sustainable development [4,5].

The responsible behavior of people during the pandemic has driven the trend of buying green products among consumers around the world [6,7,8]. Concerns for the environment and the planet’s well-being are increasingly important and relevant debates in a modern society [9], in which lifestyles and healthy behavior of adults are learned from childhood or in early adolescence [10,11]. However, it should be noted that there are sociodemographic factors such as age, gender, educational level, occupation, income level and family size that play an important role in influencing both negatively and positively the behavior of green purchasing by consumers [12,13].

Evidence shows that even low-income families are interested in consuming environmentally friendly products [14]. Therefore, applying green marketing is becoming an essential strategy for business competitiveness, since it allows companies to achieve a good reputation by involving interested parties, especially green consumers [15,16,17].

Green marketing consists of “all activities designed to generate and facilitate any exchange intended to satisfy human needs, so that the satisfaction of these needs and desires occurs, with minimal detrimental impact on the natural environment” [18]. That is to say, it is also about carrying out marketing activities without neglecting the care for the environment. In this sense, it is about promoting the consumption of green products, which refer to those products and services that support the current economic development, conserving the environment for future generations, through the use of products that do not negatively impact the environment [19]. Furthermore, consumer commitment plays a virtual role in increasing this product consumption [16,20]. In this sense, responsible companies, of all sizes, economic sectors and locations, are called upon to guide their marketing strategies to influence customers to consume more respectfully with regard to the environment [21,22,23,24].

Presently, the study related to sustainable consumption is of great interest to researchers from different parts of the world [25,26,27,28]. However, there are no studies carried out in Latin American countries; therefore, there is a need to carry out substantial research that explores the impact of environmental awareness and sustainable consumption on the environmentally responsible purchase intention of consumers in these emerging economies. Therefore, the present study is justified given that no theoretical model has been found in the literature that involves the four variables together. Given its importance for sustainability, our study intends to propose this formative model to explain environmentally responsible purchase intention in Pacific Alliance countries. On the other hand, there are sociodemographic constructions in the literature that can be used as moderator variables when analyzing the profiles of organic consumers; however, issues related to gender and place of residence have received limited attention in environmental studies [29], especially in an emerging market scenario [30]. For this reason, the present study intends to carry out an analysis considering the variables of country and gender in the relationships of the study variables, in this way, reducing this research gap.

In this sense, the study of environmentally responsible consumer behavior contributes to the literature with insights that marketers can use to design more effective green marketing strategies and develop promotional and advertising strategies that encourage responsible consumption by customers. Likewise, it is essential to study the needs and concerns of consumers in general, concerning the responsible purchasing trend that is challenging companies to transform and satisfactorily meet the new demands in the market. On the other hand, this study could be helpful to governments so that they can establish public policies to promote the production and consumption of environmentally responsible products [31,32,33,34].

The environmentally responsible purchase intention (ERPI) has been widely studied taking into account intention theories such as Reasoned Action Theory (TRA) and Planned Behavior Theory (TPB). Still, these theories do not incorporate environmental awareness and sustainable consumption as predictors of environmentally sustainable purchase intention [35,36]. ERPI is the degree of importance that a person gives to the care of their environment, and thus, with a positive attitude, it demonstrates an intention to purchase responsibly to protect the environment [7]. The research findings will contribute to the academy, providing evidence of the association of the studied constructs. This association of variables can be useful to model a theoretical framework in future research related to environmentally responsible consumer behavior and green marketing. The results of this study can also help entrepreneurs and industry managers design better business strategies and reach consumers who are demanding environmentally friendly products and services.

The structure of this article is divided as follows: After the introduction, the literature review and hypotheses are presented. Section three presents the materials and methods. Later, the fourth section presents the results and findings of the study. Finally, the fifth section shows the conclusions, limitations and future research that could be carried out to continue with this line of research.

## 2. Literature Review

The literature review is presented based on the hypotheses raised and the proposed theoretical model (see Figure 1). The rise of contemporary environmentalism dates back to the 1960s and early 1970s, as concern grew about the impact of consumption and production behavior patterns on the environment [37]. Due to the economic effect of the pandemic, consumers have become more dependent on the availability of essential consumer goods and online technology [38]. Furthermore, due to the restrictive standards of COVID-19, there has been growth in electronic commerce, which has driven consumers to seek out online retailers that provide security and convenience [5].

Customers shopping online constantly look for alternatives to compare products and prices to order online for delivery or pickup [39]. It is relevant to consider that in the post-pandemic era, consumerism and consumer behavior will continue to grow, but with a different meaning [38]. Consumption will tend toward pragmatism and purity, and consumerism will be questioned with regard to careful budgeting.

### 2.1. Purchase Intention

The intention is the cognitive representation of the willingness to adopt a particular behavior. In the context of consumer behavior, purchase intention is a prerequisite to stimulate and push consumers to buy products and services [40,41]. Theories attempt to predict the behavior of individuals based on their intention, such as the Theory of Reasoned Actions (TRA) [42], which sustains that attitude and subjective norms are antecedents that define behavior. Researchers have extensively applied the TRA to predict consumer purchase intention for multiple products and services [43,44,45,46,47]. However, another theory derived from TRA is defined as Planned Behavior Theory (TPB), which incorporates a new dimension, Perceived Behavioral Control, as a predictor of the individual’s intention [48,49]. Planned Behavior Theory, similarly to the Theory of Reasoned Action, is commonly used by researchers to predict the behavior of individuals [36,50,51,52,53,54,55]. 

### 2.2. Responsible Purchase Intention

Consumers’ attention to the environment and green products affects their purchasing decisions [34]. Deciding what type of product to buy requires being informed about the environmental situation and the benefits to conservation via responsible consumption, in terms of which products are ecological [41]. The organic purchase intention is directly and strongly influenced by perceived value, attitude and trust. On the other side, perceived behavioral control, perceived consumer effectiveness, subjective norm, perceived green quality and environmental concern are moderately related to green purchase intention [41].

### 2.3. Environment Awareness (EA) and Environmentally Responsible Purchase Intention (ERPI) during COVID-19 Pandemic

Environmental awareness refers to environmental issues and understanding the key relationships that lead to environmental impact [12]. In this vein, environmental awareness is relevant not only to individual perception and knowledge of environmental problems but also to the behavior they adopt as a result [37]. In this way, a person with low ecological awareness is less likely to buy ecological products than those consumers who are more informed about environmental problems [56]. Previous studies have shown that green awareness of environmentally responsible products such as remanufactured products, where the residual value of used products is recovered through the reuse, restoration and/or replacement of components, plays an essential role in the perceived value level of the green consumer segment [57]. Additionally, customers with a better sense of environmental responsibility are more likely to conduct an environmentally friendly activity, such as purchasing an eco-friendly vehicle [58]. Likewise, they show that environmental knowledge significantly and positively influences green purchasing intentions [59]. Furthermore, others studies indicate that environmental awareness is a crucial variable in green marketing due to its direct impact on the ecological purchase decision [16], Additionally, knowledge and awareness are the most critical factors for consumers to become environmentally responsible [60]. This means that the greater the environmental awareness, the greater the ecological consumption behavior of consumers [61]. Similarly, it has been shown that care for the environment and environmental consciousness substantially affect the desire to buy ecological products [62]. 

In recent years, the outbreak of COVID-19 has altered the perspectives of many consumers. As a result, individuals are becoming more conscious of the dangers of disregarding the planet and the environment. Even before the pandemic, there was a perception of an increase in environmental awareness and sustainability, but COVID-19 has expedited this trend and inspired more individuals to accept this duty [63,64]. According to Qian et al. [65], COVID-19 has changed individual and societal thoughts and behavior toward household food waste. While older individuals are more environmentally conscious than younger ones, the awareness level of both age groups has risen since the beginning of the epidemic. In other words, the COVID-19 epidemic favorably affected the correlation between environmental consciousness and environmentally responsible purchasing intention in Latin American nations. Therefore, people have preferred to buy ecologically friendly products to reduce environmental harm and safeguard the environment [66]. Thus, the following hypothesis is posited:

**H1:** 
*Environmental awareness (EA) impacts on environmentally responsible purchase intention (ERPI) in the context of the COVID-19 pandemic.*


### 2.4. Sustainable Consumption (SC) and Environmentally Responsible Purchase Intention (ERPI) during COVID-19 Pandemic

Consumers choose organic products as part of their responsibilities toward the environment, family and society [67]. This behavior can also be understood according to cultural, social and environmental patterns [12,41]. Sustainable consumption, also called green, ethical or environmentally responsible consumption, refers to the use of services and products that meet basic needs, provide a better quality of life, reduce environmental impact and do not compromise the needs of future generations [68]. In this sense, ecological consumers seek information and, depending on their value system, often pay more for products, driven by their willingness to participate in caring for the environment [69]. In this way, consumers collaborate with the care of the environment through ecological, decomposable, renewable, reusable and recyclable products, due to assuming a lower risk for the environment, individuals and society [41].

The existing offer, so diverse in each category, has found in ecological marketing an alternative to develop differentiating strategies. In this sense, green marketing has promoted the existence of a platform oriented toward sustainable consumption [70]. This orientation in turn has helped to increase the purchase intention and achieve a reduction in the energy consumption [71]. For example, Parashar et al. (2023) [72] mention that in India, the tendency toward consumption of organic food is closely related to awareness of health and the environment, and thus, it follows that this impacts the purchase intention.

In recent years, the COVID-19 pandemic has significantly impacted consumer habits and behaviors, resulting in a more sustainable and healthier period of consumerism [73]. For example, COVID-19 had an immediate impact on the food industry, particularly in terms of sustainable consumption and purchasing intention. The COVID-19 crisis cultivated the concept of purchasing purpose, as customers considered not just perceived attributes and buying experiences but also environmental consciousness, environmentally friendly consumption and social impact, all of which have risen during the pandemic times [64]. As a result, there has been an increase in the purchase intention of healthy and organic food [74], due to its benefits for the individual, society and the environment [75,76].

In turn, the online purchasing market has been impacted by COVID-19. So, people have expanded their internet shopping and adapted their consumption to be more environmentally friendly [77]. Some studies indicate that a way of analyzing consumer preference for green products is related to the values that guide their individual behavior [78,79]. For example, in food-related products, consumers who read food labels when shopping and think these products might have environmental or health benefits are more likely to report sustainable positive food purchase intention [80]. Thus, taking the above into account, the following hypothesis is proposed:

**H2:** 
*Sustainable consumption (SC) has an impact on environmentally responsible purchase intention (ERPI) in the context of the COVID-19 pandemic.*


### 2.5. Social Responsibility (SR) and Environmentally Responsible Purchase Intention (ERPI) during COVID-19 Pandemic

Social responsibility is considered as a guideline, a model and a precursor policy for forming people in an integral way [81]. This can be supported by training people in testimonial education, in the teaching of literature and in memory education, which can help form values [82]. 

On the other hand, from the commercial or consumer aspect, which is the scenario where people demonstrate their level of social responsibility through the choices of products or services they constantly make. Researchers have suggested that consumers with social responsibility behave in response to a formative process from their conative–volitional system, which is socially conscious; this leads to the demonstration of ethical–moral attitudes [83,84]. For example, Hoffmann-Burdzińska et al. [85] have mentioned five relevant factors of responsible consumption: knowledge of energy saving, green consumer values, social influence, beliefs and consumer awareness. In this sense, when the intensity of responsible consumption behavior is more significant, more energy-saving actions in your home or outside are contemplated. In addition, customers aware of the severity of climate change and the need to adopt corrective measures, from local and national public management to companies and citizens, are more likely to actively participate in actions in favor of caring for the environment [86]. 

According to previous studies, customers with a larger sense of environmental responsibility are more likely to undertake green behaviors, such as purchasing an eco-friendly vehicle [58]. Researchers also note that the higher the intensity of the consumer behavior determinants, the larger the number of measures made to save energy inside and outside the house [85]. Furthermore, some customers are willing to pay more for eco-friendly apparel and the resulting social approbation [8]. Nonetheless, they are not always well informed on labor exploitation problems [87]. Thus, social responsibility is claimed to influence the inclination to make ecologically responsible purchases. The following hypothesis is presented in the following manner:

**H3:** 
*Social responsibility (SR) impacts on environmentally responsible purchase intention (ERPI) in the context of the COVID-19 pandemic.*


### 2.6. Place of Residence (Country) as Moderator Variable

Sociodemographic constructs can be used as moderator variables in profiling green consumers. Among the various sociodemographic constructs, issues related to place have received limited attention in the case of environmental studies [29], especially in an emerging market environment [30]. However, there are studies that have shown that nationality is an important variable that affects consumer attitudes toward sustainability [88]. For example, in the hotel sector, it has been found that environmentally friendly hotel practices and nationality are correlated [89,90]. Location or place of residence has also been found to affect green vehicle purchase intentions in Western countries and has also shown that willingness to adopt electric vehicles is higher in urban areas [91]. One study has found that sustainable consumption intentions are highest among Chinese citizens, followed by Brazilian and Russian [92]. Another study has found that Polish and Belgian consumers differ in pro-ecological attitudes and behavior. At the level of statements, Polish consumers claim to be more concerned about ecology and declare more ecologically responsible behavior [93]. As can be seen, some studies show that the place of residence is a crucial sociodemographic variable when analyzing the intention to purchase in an environmentally responsible manner. However, it has not been possible to find this evidence in the countries of the present study; therefore, to fill this gap, this study has included the analysis of geographical moderation (country). Since it is essential to observe how this variable could influence the effect of environmental awareness, sustainable consumption, and social responsibility on environmentally responsible purchase intention during the COVID-19 pandemic.

Furthermore, according to Severo et al. [1], cultural and socioeconomic contexts should be assessed, as these factors can influence intensities. Therefore, the study assumes the effect of moderation on the country of residence of consumers. Thus, we formulate the following hypotheses:

**H4a:** 
*The consumer’s residence (country) has a moderating effect on the relationship between environmental awareness (EA) and environmentally responsible purchase intention (ERPI) in the context of the COVID-19 pandemic.*


**H4b:** 
*The consumer’s residence (country) has a moderating effect on the relationship between sustainable consumption (SC) impacts and environmentally responsible purchase intention (ERPI) in the context of the COVID-19 pandemic.*


**H4c:** 
*The consumer’s residence (country) has a moderating effect on the relationship between social responsibility (SR) and Environmentally responsible purchase intention (ERPI) in the context of the COVID-19 pandemic.*


### 2.7. Gender as a Moderating Variable

Since ancient times, gender has been considered a main variable explaining consumers’ preference for green products [12,94,95]. Likewise, it has been argued that this sociodemographic variable acts as a moderator and not necessarily as a determinant of the consumer’s green preference [96]. This is justified because men and women behave and socialize differently [97,98]. Gender socialization theory holds that girls and boys develop different social expectations and values, because they go through other socialization processes from early childhood [99]. Research has shown that women are more concerned than men about environmental problems and therefore have a more positive attitude toward products that protect the environment [30,100]. Women have been found to possess a more robust environmental attitude than men in different countries [29,101]. Regarding gender differences regarding sustainable consumption and attitudes toward the environment are not universal or conclusive. However, several studies have revealed that women are more likely than men to engage in green purchasing behavior [12,56]. Taking the above into account, this study has included the analysis of gender moderation, as a variable that could influence the effect of environmental awareness, sustainable consumption and social responsibility with an environmentally responsible purchase intention during the COVID-19 pandemic. This mode presents the following hypotheses:

**H5a:** 
*The gender of the consumer (country) has a moderating effect on the relationship between environmental awareness (EA) and environmentally responsible purchase intention (ERPI) in the context of the COVID-19 pandemic.*


**H5b:** 
*The gender of the consumer (country) has a moderating effect on the relationship between sustainable consumption (SC) and environmentally responsible purchase intention (ERPI) in the context of the COVID-19 pandemic.*


**H5c:** 
*The gender of the consumer (country) has a moderating effect on the relationship between social responsibility (SR) and environmentally responsible purchase intention (ERPI) in the context of the COVID-19 pandemic.*


## 3. Materials and Methods

### 3.1. Context and Method

This article aims to examine the influence of environmental awareness, sustainable consumption and social responsibility on consumers‘ environmentally responsible purchase intention in the Pacific Alliance countries, that is, Chile, Colombia, Mexico and Peru. Likewise, this work analyzes if there is a moderating effect related to geography and the gender of the consumer.

The study was carried out under a quantitative approach and a non-cross-sectional design through a self-administered questionnaire [102]. The survey was applied in the countries of the Pacific Alliance that are Spanish speaking. Therefore, it was necessary to use the back-translation method proposed by Brislin [103].

### 3.2. Sample and Procedure

For the data collection of the present investigation, non-probability sampling was applied for convenience [104], through an online survey through the Google form, whose link was shared through social networks, such as Facebook, Instagram, LinkedIn, email and WhatsApp. In addition, the use of social networks and the Internet contributed to the randomness and diversity of the characteristics of the respondents [1]. The survey was applied during the months of July to September 2021.

For respondents to participate in the survey, they were informed that their participation was voluntary, that the data collected would be analyzed anonymously and that they would be used exclusively for academic purposes. In this way, it was possible to recover 1646 surveys corresponding to consumers from the four countries that responded to the survey. The sample by country was 24.4% from Peru (n = 402), 25.4% from Mexico (n = 418), 26.1% from Colombia (n = 401) and 24.1% from Chile (n = 397). The sample is equitable among each country; therefore, the size of the research sample size meets the requirements [102] (see Table 1).

Women are predominant in the sample data. There were 963 female participants (58.5%), followed by 683 male participants (41.5%). The average age was 33.10 years (SD = 12.20). The average age of the men was 33.44 (SD = 12.58), and the average age of the women was 32.85 (DE = 11.93). Table 1 shows the sociodemographic data of the sample of this study.

### 3.3. Measures

To build this research model, the items were adopted from previous studies. To assess environmental awareness, sustainable consumption and social responsibility of consumers, the instrument developed by Severo [1] was taken into consideration, and to determine the environmentally responsible purchase intention, the construct developed by Kumar [8] was taken into account. For the semantic validation of the questionnaire, two focus group sessions were held with the participation of eight academics from different areas, specializing in marketing, business management and sustainability, who evaluated the questionnaire to verify the scales and items.

The final questionnaire was divided into two parts. The first section featured 19 claims related to the effects of COVID-19 on environmentally responsible purchase intention. Specifically, this section was separated into four sections according to the variables analyzed. The constructs are: environmental awareness (EA), sustainable consumption (SC), social responsibility (SR) and environmentally responsible purchase intention (ERPI). The second section was related to demographic data, such as country, age, and gender.

Each item was written as a statement to be evaluated (Table 2), applying a five-point Likert-type scale. The values were: 1 (totally disagree), 2 (disagree), 3 (neither agree nor disagree), 4 (agree) and 5 (totally agree). In this sense, each number and item was explained to all the interviewed participants so that they understood and responded appropriately.

In the present study, it was necessary to eliminate the items EA1, EA2, EA3, EA4, SR1 and SR4, in order to achieve a good fit of the model, since the fit indices of the initial model (model 1) were low; therefore, the model was re-specified based on the modification index (MI) and the standardized residue matrix [105]. In this sense, it was observed that the items EA1, EA2, EA3, EA4, SR1 and SR4 had higher MI values in the correlation with other measurement errors and in the standardized residue matrix. These items exceeded the value of +/− 2.58 [106]. In this way, the analysis was carried out without items EA1, EA2, EA3, EA4, SR1 and SR4 (Model 2), obtaining adequate fit indices (see Table 3).

According to the research of Severo, the proposed variables can be improved by excluding factors that contribute little to the explanation of construct variability or by adding new observable variables [1]. In this sense, in the current investigation, some items of the original construct have been removed. Specifically, items of the environmental awareness (EA1, EA2, EA3 and EA4) and social responsibility (SR1 and SR4) variables have been deleted from items to improve the normality and reliability tests and the original study’s fit model. Specifically, Cronbach’s Alpha (CA), Composite Reliability (CR) and Average Variance Extracted (AVE) have improved. In the original study, the variables environmental awareness (CA: 0.826; CR: 0.888; AVE: 0.727) and social responsibility (CA: 0.745; CR: 0.826; AVE: 0.501) are adequate. However, our results have improved the normality and reliability tests; environmental awareness (CA: 0.830; CR: 0.846; AVE: 0.736) and social responsibility (CA: 0.864; CR: 0.845; AVE: 0.646). In addition, the original study’s fit model has lower values than ours (RMSEA = 0.085; CFI = 0.827). Our results have the best model fit (RMSEA = 0.047; CFI = 0.989). Therefore, we consider that our exclusion of items in the constructors positively affects the impact on the validity and reliability of the results.

From a theoretical point of view, the removed items are connected to environmental awareness actions, for instance, the growth in recycling and separation of organic waste, the reduction in water as a finite resource, apprehension about the future of natural resources, and the relevance of decreasing environmental pollution [1]. In this sense, we could say that the COVID-19 pandemic has increased environmental awareness in the countries of the Pacific Alliance. However, these nations are still in the phase of detecting a problem but have not yet taken measures to prevent it.

### 3.4. Statistical Analysis

The first statistical process evaluated the reliability and validity of the model. Specifically, this study used the Cronbach’s Alpha method to measure the latent variables’ reliability and the items´ internal consistency. Then, a confirmatory factor analysis (CFA) was applied to verify the fit and measurement of the model. The research used two statistical programs. First, the IBM SPSS Statistics software was used for the descriptive analysis. Secondly, the AMOS Software was selected to check the convergent and discriminant validity and to test the hypotheses through a multigroup Structural Equation Modeling (SEM).

SEM is considered a suitable method for this type of investigation. First, this method is highly recommended for evaluating cause–effect relationships in descriptive models [104]. Second, SEM is a perfect method to test the hypothesis of dependency relationships, correlations and the effects of moderator variables [107]. Finally, recent studies applied SEM to analyze and demonstrate robustness in measurements and structural evaluation [4,10,108,109].

## 4. Results

In this section, we present the results of the reliability and validity analysis (convergent and discriminant) and the SEM estimates of the hypotheses’ general proposals. Therefore, the results were performed for the entire sample as a whole, that is, for the total sample of 1646, in addition to an analysis of invariance and effects moderators, for which the subsamples were used for grouping both by country and gender.

### 4.1. Reliability and Validity Analysis

To evaluate the reliability of the instruments, Cronbach’s Alpha (CA) has been used, which is an indicator most used for the verification of the scales [110]. A level considered adequate is values of the latent variables greater than 0.70 [104]. The results of this study are satisfactory since all the indicators are above 0.830 (see Table 4). Each latent variable’s observed factor loading (Std Beta) is between 0.612 and 0.967 The above values comply with Fornell and Larcker’s [111] index requirements. 

Regarding convergent validity, the mean variance extracted (AVE) and Composition Reliability (CR) were used. The AVE indicator is considered acceptable with values equal to or greater than 0.5 [104]. CR must be greater than 0.6 [112]. In this research, each latent variable shows a good level: AVE, with values greater than 0.629, and CR, with values equal to or greater than 0.845.

Fornell and Larcker’s criteria [111] and the proportion of heterotrait–monotrait correlations for discriminant validity [113]. Once again, the results establish an adequate validity of the proposed model (see Table 5 and Table 6).

Table 7 shows the indicators of the adjustment of the measurement model of the model scale under study, and it is observed that it meets all the indicators adequately.

### 4.2. SEM Estimates of the Proposed General Hypotheses

The results have indicated that the measurement model provided good model fit values. Specifically, with χ^2^⁄df = 4.690 (267.318/57,000), an acceptable value is between 3 and 5 [67]. In the case of incremental values, a good fit will be considered with values: CFI (Comparative Fit Index) > 0.95 [114]. The parsimony indices fit the valued model well: RMSEA (Root-Mean-Square Error of Approximation) < 0.06 [114], SRMR (Standardized Residual Root Mean Squared) < 0.08 and the PClose (*p* of close Fit) > 0.05 [114]. The results have indicated a satisfactory goodness of fit: CFI = 0.989, SRMR = 0.033, RMSEA = 0.047, and PClose = 0.767.

According to the influence of environmental awareness (EA) on the environmentally responsible purchase intention (ERPI), the study reveals a positive and significant relationship (EA → ERPI; CR = 4648 ***; SE = 0.032). Regarding the influence of sustainable consumption (SC) on the environmentally responsible purchase intention (ERPI), the study reveals a positive and significant relationship (SC → ERPI; CR = 14,493 ***; SE = 0.904), and in terms of the influence of social responsibility (SR) on the environmentally responsible purchase intention (ERPI), the study reveals a positive and significant relationship (SR → ERPI; CR = 4017 ***; SE = 0.029) Consequently, the three hypotheses H1, H2 and H3 are supported by this research (see Table 8 and Figure 2).

### 4.3. Analysis of Invariance and Moderating Effects

A moderation analysis has been carried out to observe differences between countries and gender (see Table 9). The results show variations between the variables related to the magnitude and the significance effect.

Configural and metric invariance tests have been carried out to verify the origin of these differences for the variables: country and gender (see Table 10 and Table 11). Invariance is defined as the verification process. The configural variance is confirmed when the indicators of the fit of the model continue to be good after having carried out the division of the groups. In this sense, in Table 9, the indicators (CMIN/DF; CFI, SRMR, RMSEA and PClose) present an excellent model fit, and therefore, the configural invariance is verified.

Table 11 shows the metric invariance according to Cheung and Rensvold [115]; if the comparative index between the CFIs of the two compared models is less than 0.01, it is affirmed that invariance exists, and therefore, the SEM is performed to validate the relationship hypotheses. This analysis indicates that both the country and gender variables present invariance. Therefore, the groups are not different at the model level for the variables country and sex, the difference being relevant at the road level.

It is verified that the difference between CFI must be less than 0.01. Since the difference in the CFI is less, they are almost equal. Thus, it satisfies the invariance of covariances.

## 5. Discussion

This research was carried out to empirically evaluate the influence of environmental awareness (EA), sustainable consumption (SC) and social responsibility (SR) on the environmentally responsible purchase intention (ERPI) of a sample of consumers equally distributed in each country. From the Pacific Alliance, 402 participants were from Peru (24.4%), 418 from Mexico (25.4%), 401 from Colombia (26.1%) and 397 from Chile (24.1%) with a total of 1646 valid responses. After performing the data analysis through Structural Equation Modeling (SEM), the results have confirmed the influence of AC, SC and SR on the ERPI.

This study has confirmed the influence of environmental awareness (EC) on consumers’ sustainable purchase intention (ERPI) in Peru, Mexico, Colombia and Chile. This result is in line with previous studies [116], which point out that environmental awareness (EA) is an essential motivator for developing behavioral intention. Likewise, this result is contrasted with the findings of [59], in which the authors point out that environmental knowledge significantly and positively influences green purchasing intentions. Our results show that this influence is direct and significant, following what was indicated by [61], which affirms that the greater the environmental awareness, the greater the ecological consumption behavior of consumers. This finding is important because it can be stated that if the consumer has high environmental awareness, they will be more willing to pay more for environmentally friendly products, since eco-friendly products are generally more expensive than products that are not eco-friendly. Therefore, companies are not willing to produce this type of product considering that consumers are not willing to pay the price of this type of product [58,117]. Given that the present study was carried out in the context of the COVID-19 pandemic, it is essential to emphasize that the results show that this scenario has had a positive impact on the relationship between environmental awareness (EA) and environmentally responsible purchase intention (ERPI) in all the countries studied, both in men and women, confirming studies that indicate that this event has had an effect on consumers [118,119]. This could mean an opportunity for companies in this region to propose new alternatives in sustainable products in their portfolios, adjusting to the needs of customers to achieve sales effectively that generate value for this consumer market [120].

The findings suggest that sustainable consumption (SC) is the most important factor influencing environmentally responsible purchase intention (ERPI) due to its higher path coefficient than CE and SR. This finding suggests that consumers who have a higher sustainable consumption are more likely to adopt an environmentally responsible purchase intention, for example, to buy remanufactured products [57], be willing to pay a higher price for an environmentally friendly hotel service [117] or buy a car that does not damage the environment. This means that sustainable consumption (SC) has made an important contribution in recent times in society, since consumers and customers have become aware of how important for the environment it is to buy environmentally friendly products, as studies such as [82,120] have shown. In the case of the Pacific Alliance countries, this study has demonstrated that by purchasing environmentally responsible products, a better culture of support for companies is built, and the consolidation of strategies that add value to brands for improved reputation is achieved, as concluded in prior studies [121,122]. This is how when mentioning sustainable consumption (SC), factors such as gender and demographic location, which influence the purchase decision, are taken into account. At the same time, the perceptions of individuals on the environmental issue contribute to and are strategic for sustainable consumption, thus representing a competitive advantage in companies, and generating the environmentally responsible purchase intention (ERPI) of individuals in the different Latin American countries analyzed.

Along the same lines, the hypothesis was confirmed that, in the context of the COVID-19 pandemic, consumer social responsibility (SR) has positively impacted on their environmentally responsible purchasing intention (ERPI). This supports previous research evidence on social responsibility and consumer intent to protect the environment during the COVID-19 pandemic [1,108]. Furthermore, there are studies [123] that point to the commitment of consumers toward the activities of corporate social responsibility of companies (CSR) and how this contributes to their purchase intention. That is, consumers’ growing concern for the environment puts pressure on organizations to be environmentally responsible [124]. Therefore, it is important that business decision makers take into account that consumers like to associate with socially responsible companies because the social values of the consumer and those of the company overlap each other [125]; thus, this constitutes an opportunity for companies seeking to position themselves in the market and achieve customer loyalty. In this sense, a company committed to the environment and society will be more attractive to customers, who will consider it more trustworthy and will have positive feelings toward the company [126,127,128]. In this sense, Carroll [129] points out that the philanthropic activities of companies are generally part of society’s expectations. That is, companies are expected to operate ethically. This means that the company has an obligation to do what is right, fair and equitable and to avoid or minimize harm to all stakeholders with whom it interacts. In other words, the company is expected to be a good corporate citizen, giving back and contributing financial, physical and human resources to the communities of which it is a part. Thus, studies on the effect of corporate social responsibility (CSR) on consumer buying behavior indicate that being legally responsible is not enough for stakeholders, as organizations are expected to not only conform to the law but also be ethically responsible [130]. Therefore, the existence of a positive relationship between the CSR activities of a company and consumer behavior toward the company and its products must be kept in mind [124,131].

A significant finding of the present study is the empirical evidence of the moderating effect of gender and country on the influences of the EA, the SC and the SR on the environmentally responsible purchase intention (ERPI). Regarding the moderating effect of customers’ country of residence, our results indicate that consumers from Peru and Colombia contrast significantly with regard to the three hypotheses; however, in the case of Mexico, the SR does not significantly influence the ERPI, while in the case of Chile, environmental awareness does not have a significant influence on the ERPI. This finding is supported by the literature, which indicates that there are factors such as gender, age, income, education, income and family size that influence consumers’ green purchase intention [12,95]. In this scenario, companies must diversify their strategies to promote environmentally responsible purchasing behavior considering how different nationalities address the need for sustainability in purchasing processes and understanding that there is a different understanding and awareness of the environmental urgency on the planet.

Regarding the moderating effect of consumer gender, our results indicate that EA, SC and SR directly influence the ERPI of women in the four countries of the Pacific Alliance. However, these hypotheses are not fully met in the case of men; that is, our results show that in the case of men, SR does not significantly influence ERPI. This finding is contrasted by [12,56], in which the authors point out that gender differences regarding sustainable consumption and environment attitudes are not universal or conclusive; however, several studies have found that women are more likely than men to participate in a green buying behavior.

It can be affirmed that sustainable consumption has become a priority for many countries, which observe environmental development and prospect it with interventions that allow humans to live better in society; thus, they adapt the markets with a consumption model that allows them to act responsibly [132]. In recent years, the COVID-19 pandemic has contributed to a better awareness of products that help the environment and the sustainability of the planet in their clean processes [82], allowing companies to propose new alternatives in healthy products in their stores. Portfolios adjust to the clients’ need to achieve the sale effectively with a generator of clear value in the market [120]. 

The era of sustainable products has impacted various sectors and the way in which they develop their production schemes, which gives the customer a better awareness in the responsible purchase of products that help the environment [121], thus consolidating culture in society. In addition, this allows companies to build an adequate brand reputation and become aspirational in the market [122] for the human talent that is part of the companies’ production processes, by providing motivating and consolidating work schemes that provide well-being and satisfaction to those who are part of the company [13], to build trust and achieve better articulation with business policies that support employees and customers [133].

### Implications

This research has implications for managerial decision making in the public and private sectors. Our findings provide evidence that environmental awareness, sustainable consumption, and social responsibility have a direct influence on environmentally responsible purchase intention of consumers from countries of the Pacific Alliance during the COVID-19 pandemic.

Firstly, environmental awareness has increased significantly during the COVID-19 pandemic [1], manifested in consumers who prefer sustainable products that are less harmful to the environment. Therefore, companies must adopt sustainable and green business practices to align with individual customer behavior. For instance, companies can educate the population through recycling activities and reducing plastic packaging [99]. In the case of the public sector, the government should advertise how and when a product or service is friendly to the environment and society, thus getting closer to its citizens’ environmental awareness [108]. Likewise, governments must develop public policies that help transform a more sustainable world, for example, by proposing public programs to reduce pollution, encourage recycling and promote efficient water consumption [134].

Secondly, this research indicates that sustainable consumption has revolutionized the world during the pandemic. Consumers have changed the consumption paradigm toward a friendly orientation toward the environment. From this perspective, managers and entrepreneurs should focus business strategies on their products and services and their production, distribution, and sales channels [135]. However, consumers have inspected the intention to consume friendly products, but the COVID-19 pandemic has hit the world economy hard. For this reason, governments must consider public policies to stabilize and reactivate the market [136].

Finally, this research affirms that consumers in the countries of the Pacific Alliance are developing greater environmental awareness, but it is still necessary to develop their consumption habits to take care of the environment by reducing waste through prevention, reuse, recycling and water care. For this reason, organizations, in addition to producing environmentally friendly products and reducing pollution and greenhouse gas emissions [108,124], must also apply communication strategies to promote more environmentally friendly behavior by people. Additionally, the government must create public policies to ensure that organizations reduce their indicators of environmental contamination and increase control over environmental damage.

## 6. Conclusions

Unlike existing research in a non-COVID-19 pandemic context on environmentally responsible purchasing intention (ERPI) factors [9,14,16,17], our research proposes a research framework in Latin America focused during the COVID-19 pandemic, specifically on consumer behavior in terms of environmental awareness (EC), sustainable consumption (SC), social responsibility (SR), and environmentally responsible purchase intention (ERPI) in the context of the COVID-19 pandemic.

This study contributes to a better understanding of the impact of the pandemic on the environmentally responsible purchasing behavior of consumers in the Pacific Alliance countries, comprising Peru, Mexico, Colombia and Chile. However, since the study was carried out during the year 2021, a period in which the pandemic was spreading throughout the world and the entire society lived in a situation of fear and uncertainty, it is not possible to generalize the findings for the post-pandemic period. When the pandemic has already been controlled or has ended, it is therefore recommended to continue carrying out similar investigations to know the evolution of this behavior of the consumers of the Pacific Alliance. In this sense, it is necessary for future research to examine whether consumers’ eco-friendly and pro-environmental behavior is maintained over time or is just a temporary event. It must also be considered that this study has only been carried out in Peru, Mexico, Colombia and Chile. Therefore, future research should be carried out where other Latin American countries can be included, providing a better understanding of this phenomenon.

In conclusion, the participants of the countries of the Pacific Alliance studied in this research, namely, Peru, Colombia, Chile and Mexico, recognize the importance of purchasing products that are friendly to the environment; at the same time, females have the greatest awareness of now being the time to analyze this issue of responsible and environmentally friendly consumption, being willing to make a purchase decision and acquire products that provide sustainability to health and the environment, as a better alternative for the future. Among the most representative factors, awareness is perceived on issues of sustainable consumption and environmental responsibility during and after the COVID-19 pandemic, a topic that facilitates structuring new business models to provide better products and services to the national and international market. In the same way, it is important to identify tastes, desires and purchasing preferences to reformulate eco-friendly portfolios with the environment and with society.

This research had limitations such as the use of convenience sampling. Although it had enough participants, it is recommended for future studies that more robust data collection procedures be used. Another aspect to consider as a limitation is that even though a similar size of the sample size has been achieved in the four countries, it may not have the same predictive capacity due to the difference in consumers in each country. That is why it is recommended to carry out further studies considering the sample size according to the size of consumers in each country. Likewise, future studies are recommended considering other sociodemographic variables such as education and income levels, to determine if they could influence more environmentally responsible consumption.

Despite the limitations indicated, the main contributions of this work consist in revealing the positive influence of sustainable consumption, providing some guidelines for practical applications and the opportunity to apply this research framework in different contexts of the given place and time, since the results of this study were obtained during the COVID-19 pandemic.

## Figures and Tables

**Figure 1 behavsci-13-00221-f001:**
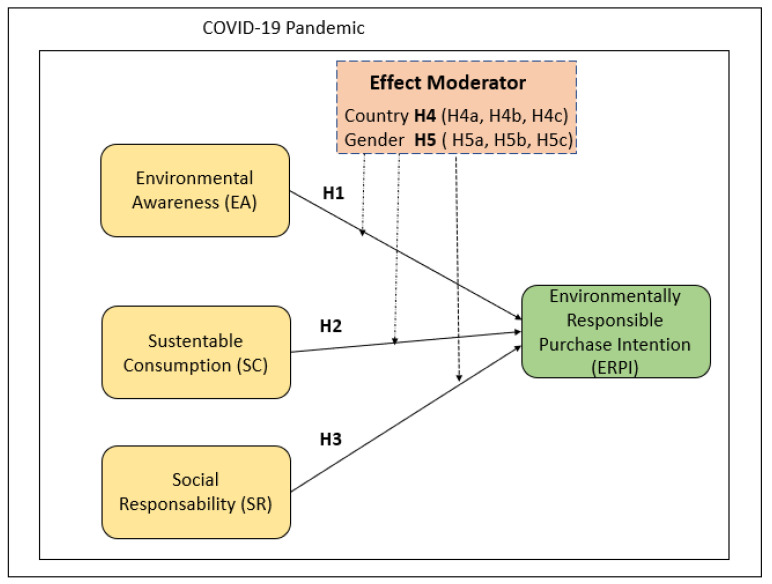
Framework and integrated model.

**Figure 2 behavsci-13-00221-f002:**
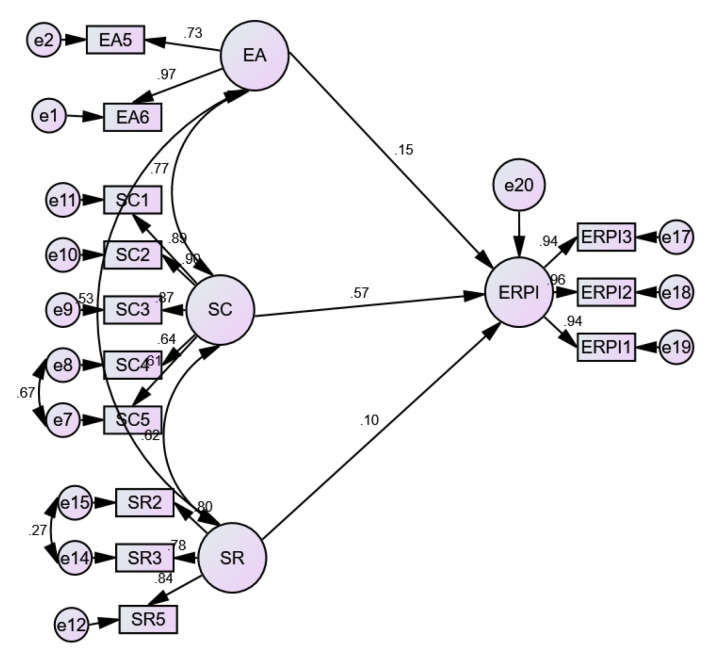
Structural model.

**Table 1 behavsci-13-00221-t001:** Sociodemographic data of the sample by gender and country of origin.

		Country				Total
		Peru	Mexico	Colombia	Chile	
Men	Count	162	163	159	199	683
	% of the total	9.8%	9.9%	9.7%	12.1%	41.5%
Woman	Count	240	255	270	198	963
	% of the total	14.6%	15.5%	16.4%	12.0%	58.5%
Total	Count	402	418	429	397	1646
	% of the total	24.4%	25.4%	26.1%	24.1%	100.0%

**Table 2 behavsci-13-00221-t002:** Construct.

Variable	Code	Measurement Items in English	Measurement Items in Spanish
Environmental awareness (EA)		The COVID-19 pandemic…	La pandemia COVID-19…
	EA1	It has made me increase the separation of organic and recyclable waste.	Me ha hecho incrementar la separación de residuos orgánicos y reciclables.
	EA2	It has caused me to reduce the consumption of water because it is a limited resource.	Me ha provocado reducir el consumo de agua porque es un recurso limitado.
	EA3	It made me care more about natural resources for future generations.	Me hizo preocupar más por los recursos naturales para las generaciones futuras.
	EA4	It made me realize that air pollution was reduced.	Me hizo dar cuenta que se redujo la contaminación del aire.
	EA5	It made me realize the negative environmental impact caused to the planet.	Me hizo dar cuenta del impacto ambiental negativo causado al planeta.
	EA6	It has increased my environmental awareness.	Ha aumentado mi conciencia ambiental.
Social Consumption (SC)		The COVID-19 pandemic…	La pandemia COVID-19…
	SC1	It made me change my consumption habits to take care of the environment.	Me hizo cambiar mis hábitos de consumo para cuidar el medio ambiente.
	SC2	It made me buy environmentally friendly products.	Me hizo comprar productos amigables con el medio ambiente.
	SC3	It led me to reduce the generation of waste through prevention, reuse and recycling.	Me llevó a reducir la generación de desechos mediante la prevención, la reutilización y el reciclaje.
	SC4	It has reduced the impact on climate change, reducing greenhouse gases.	Ha reducido el impacto en el cambio climático, disminuyendo gases de efecto invernadero.
	SC5	It has reduced the damage to forests and, in general, to the ecosystem.	Ha reducido el daño a los bosques y, en general, al ecosistema.
Social responsability		The COVID-19 pandemic…	La pandemia COVID-19…
	SR1	It has made me more sensitive to social issues.	Me ha vuelto más sensible a los problemas sociales.
	SR2	It made me donate food or clothes.	Me hizo donar comida o ropa.
	SR3	It made me donate money to people or institutions in need.	Me hizo donar dinero a personas o instituciones necesitadas.
	SR4	It made me consume products/services from companies that are socially and environmentally committed.	Me hizo consumir productos/servicios de empresas comprometidas social y medioambientalmente.
	SR5	You are helping to support socially vulnerable people.	Está contribuyendo a apoyar a personas socialmente vulnerables.
Environmentally responsible purchase intention		The COVID-19 pandemic has made me…	La pandemia COVID-19 ha hecho que yo…
	ERPI1	You are planning to buy green products in the future.	Esté planeando comprar productos ecológicos en el futuro.
	ERPI2	You are planning to buy organic products regularly.	Esté planeando comprar productos ecológicos con regularidad.
	ERPI3	Decide to spend more effort in consuming organic products compared to traditional ones.	Decida dedicar mayor esfuerzo en consumir productos ecológicos en comparación con los tradicionales.

**Table 3 behavsci-13-00221-t003:** Indices of statistical goodness of fit of the measurement model (n = 1646).

Fit indices		Model 1		Model 2	
Measure	Threshold	Estimate	Interpretation	Estimate	Interpretation
CMIN	--	2,125,303	--	267,318	--
DF	--	138,000	--	57,000	--
CMIN/DF	Between 1 and 3	15,401	Terrible	4690	Acceptable
CFI	>0.95	0.924	Acceptable	0.989	Excellent
SRMR	<0.08	0.060	Excellent	0.033	Excellent
RMSEA	<0.06	0.094	Terrible	0.047	Excellent
PClose	>0.05	0.000	Not Estimated	0.767	Excellent

**Table 4 behavsci-13-00221-t004:** Scale items, factor loadings, composite reliabilities and average variance extracted.

Constructs	Items	Std Beta	CA	CR	AVE
Environmental awareness (EA)	EA5	0.734 ***	0.830	0.846	0.736
	EA6	0.967 ***			
Social consumption (SC)	SC1	0.889 ***	0.901	0.892	0.629
	SC2	0.903 ***			
	SC3	0.871 ***			
	SC4	0.637 ***			
	SC5	0.612 ***			
Social responsibility (SR)	SR2	0.797 ***	0.864	0.845	0.646
	SR3	0.777 ***			
	SR5	0.835 ***			
Environmentally responsible purchase intention (ERPI)	ERPI1	0.944 ***	0.964	0.964	0.898
	ERPI2	0.956 ***			
	ERPI3	0.944 ***			

Source: Self-elaboration. Note: CA = Cronbach’s Alpha; CR = Composite Reliability; AVE = Average Variance Extracted. *p*-value = *** *p* < 0.01.

**Table 5 behavsci-13-00221-t005:** Fornell and Lacker’s criteria for discriminant validity.

	CR	AVE	EA	SC	SR	ERPI
EA	0.846	0.736	0.858			
SC	0.892	0.629	0.768 ***	0.793		
SR	0.845	0.646	0.529 ***	0.621 ***	0.804	
ERPI	0.964	0.898	0.641	0.750	0.538	0.948

Source: Self-elaboration. *p*-value = *** *p* < 0.01.

**Table 6 behavsci-13-00221-t006:** Heterotrait–monotrait ratio for discriminant validity.

	EA	SC	SR	ERPI
EA				
SC	0.798			
SR	0.534	0.624		
ERPI	0.649	0.727	0.520	

Source: Self-made.

**Table 7 behavsci-13-00221-t007:** Adjustment of the General Model.

Measure	Estimate	Threshold	Interpretation
CMIN	267,318	--	--
DF	57,000	--	--
CMIN/DF	4690	Between 1 and 3	acceptable
CFI	0.989	> 0.95	excellent
SRMR	0.033	< 0.08	excellent
RMSEA	0.047	< 0.06	excellent
PClose	0.767	> 0.05	excellent

Source: Own elaboration. Hu & Bentler [114] recommend a combination of CFI > 0.95 and SRMR < 0.08. To further solidify the evidence, add the RMSEA < 0.06. CFA = Confirmatory factor analysis; RMSEA = Root-mean-square error of approximation; PClose = p for close.

**Table 8 behavsci-13-00221-t008:** SEM Estimates on Hypothesis Tests.

H	Hypothesis			Estimate	S.E.	C.R.	*p*	Decision
H1	EA	------>	ERPI	0.147	0.032	4648	***	supported
H2	SC	------>	ERPI	0.904	0.062	14,493	***	supported
H3	SR	------>	ERPI	0.117	0.029	4017	***	supported

Source: Self-elaboration. Note: EA = environmental awareness; SC = sustainable consumption; ERPI = environmentally responsible purchase intention; C.R = Composite Reliability; *p*-value = *** *p* < 0.01.

**Table 9 behavsci-13-00221-t009:** Moderating effect of country and gender by multigroup analysis.

	Country		SP			Estimate	S.E.	C.R.	*p*	Decision
Country	Peru	H1	AE	--->	ERPI	0.197	0.067	2927	0.003	supported
		H2	SC	--->	ERPI	0.718	0.125	5751	***	supported
		H3	MR	--->	ERPI	0.127	0.063	2028	0.043	supported
	Mexico	H1	AE	--->	ERPI	0.195	0.066	2942	0.003	supported
		H2	SC	--->	ERPI	0.913	0.132	6937	***	supported
		H3	MR	--->	ERPI	0.069	0.055	1255	0.209	Not supported
	Colombia	H1	AE	--->	ERPI	0.207	0.056	3672	***	supported
		H2	SC	--->	ERPI	0.732	0.096	7600	***	supported
		H3	MR	--->	ERPI	0.117	0.053	2208	0.027	supported
	Chili	H1	AE	--->	ERPI	0.017	0.072	0.234	0.815	Not supported
		H2	SC	--->	ERPI	1395	0.201	6924	***	supported
		H3	MR	--->	ERPI	0.176	0.064	2745	0.006	supported
Gender	Men	H1	AE	--->	ERPI	0.174	0.047	3687	***	supported
		H2	SC	--->	ERPI	0.688	0.069	10,005	***	supported
		H3	MR	--->	ERPI	0.036	0.049	0.721	0.471	not supported
	Women	H1	AE	--->	ERPI	0.103	0.043	2393	0.017	supported
		H2	SC	--->	ERPI	0.619	0.048	12,805	***	supported
		H3	MR	--->	ERPI	0.173	0.037	4625	***	supported

Source: Self-made. Note: H = hypothesis; SP = structural road; E = estimated; EA = environmental awareness; SC = sustainable consumption; SR = social responsibility; ERPI = environmentally responsible purchase intention; *p*-value = *** *p* < 0.01; SE = standard error; CR = Composite Reliability.

**Table 10 behavsci-13-00221-t010:** Configural invariance.

Measure	Threshold	Country		Gender	
		Estimate	Interpretation	Estimate	Interpretation
CMIN	--	614,855	--	373,860	--
DF	--	228,000	--	114,000	--
CMIN/DF	Between 1 and 3	2697	excellent	3279	acceptable
IFC	>0.95	0.979	excellent	0.986	excellent
SRMR	<0.08	0.047	excellent	0.041	excellent
RMSEA	<0.06	0.032	excellent	0.037	excellent
PClose	>0.05	1000	excellent	1000	excellent

**Table 11 behavsci-13-00221-t011:** Metric Invariance.

Variable	Model	NFI	R.F.I.	IFI	TLI	IFC
		Delta1	rho1	Delta2	rho2	
Country	Unconstrained	0.968	0.956	0.979	0.972	0.979
	Measurement weights	0.966	0.958	0.979	0.974	0.979
	Structural covariances	0.962	0.959	0.977	0.975	0.977
	Measurement residuals	0.948	0.950	0.964	0.966	0.964
	Saturated model	1000		1000		1000
	Independence model	0.000	0.000	0.000	0.000	0.000
Gender	Unconstrained	0.980	0.973	0.986	0.981	0.986
	Measurement weights	0.979	0.974	0.986	0.982	0.986
	Measurement intercepts	0.976	0.972	0.983	0.980	0.983
	Structural covariances	0.975	0.973	0.982	0.981	0.982
	Measurement residuals	0.972	0.973	0.981	0.981	0.981
	Saturated model	1000		1000		1000
	Independence model	0.000	0.000	0.000	0.000	0.000

## Data Availability

Data availability can be requested by writing to the corresponding author of this publication.

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
