# Peer review of "Environmentally Responsible Purchase Intention in Pacific Alliance Countries: Geographic and Gender Evidence in the Context of the COVID-19 Pandemic"

_behavsci, 2023, doi:10.3390/bs13030221_

Round 1
Reviewer 1 Report
The paper provides a contribution to both theoretical and empirical knowledge base and is within the scope of the journal. However, the author(s) should consider several suggestions for paper improvement and scientific clarity (suggestions are listed in the order they were encountered in the manuscript):
1) In the paper title, the name of the virus and related disease should be written in capital letters (COVID-19)
2) Due to several instances of unusual linguistic and/or phrasal choices, additional proofreading of the manuscript is advised.
3) In the Abstract section, the authors state: “Likewise, analyze if there is a moderating effect related to geography and the gender of the consumer”. This statement should be rephrased as it is unusually structured.
4) In section 1.1.3. the authors use the term “remanufactured product”. Is this the synonym for a recycled product, or is there a difference? Additional elaboration would be beneficial.
5) Manuscript guidelines at the beginning of section 2 should be omitted.
6) The authors should avoid repeating the sentences from the Abstract in the manuscript.
7) Table 3 provides only a portion of the used items in corresponding constructs. For example, only 2 items are presented for Environmental awareness, while table 2 suggests 6 items. The proposed structural model also focuses on a smaller number of items. Additional clarification of the missing items is absolutely necessary.
8) Research limitations should be addressed and elaborated on in more detail.
Author Response
Dear Reviewers,
Thank you very much for your informed comments, which helped us so much in improving the manuscript. We appreciated the time you spent in doing this and tried our best to address all your comments.
We hope that this revised version of the paper reaches the expected standard, worthy of publication in this journal.
A detailed list of answers to your comments and suggestions is reported below.
Many thanks for your time.
Best regards,

Reviewer 2 Report
This is an interesting and challenging paper. The research is addressed to explore the influence of environmental awareness, sustainable consumption and social responsibility on the intention of an environmentally responsible consumption. In my opinion, the paper is well written and the objectives of the research perfectly fit with the aim of the journal.
The research design and methods are fully appropriate and the researchers find significant results. The sample is quite representative, consisting of 1,646 individuals distributed among four Latin-American countries (Chile, Colombia, Mexico and Peru). Although the distribution of surveyed consumers is almost equal between these countries and the sampling error is similar due to the size of population (400 individuals for infinite universes, provide an acceptable sampling error according to statistical methods), the authors should be cautious at the time of analyzing results at the national level, because of the huge differences in the number of consumers. Similar size of sample in Mexico and Chile do not necessarily provide the same predictive capacity, particularly when you use the country of residence as a moderator. Be careful on that point, please.
The methodology is based on quantitative research methods and, in particular, on the implementation of structural equations modelling. This technique is proper for the measurement and analysis of the relationships between observed and latent variables. The usage of the suggested SEM analysis perfectly fits with the aim of the research and the sampling process. The inclusion in the paper of a detailed description of how the variables have been designed and which questions in the survey were used to construct the different variables as well as the reliability analysis is highly recommendable. Consequently, the authors find some clear and sound linear causal relationships among variables, providing some interesting results.
In my opinion, the main contributions of this paper are to reveal the positive influence of sustainable consumption, the supply of some guidelines for practical applications and the opportunity to apply this research framework in different place and time contexts because, as the authors denote, the results were obtained during the Covid-19 pandemic.
I would only wonder why other socioeconomic attributes of survey respondents have not been taken into consideration, as the education or the income levels of individuals, for example. It could influence the intention to a more environmentally responsible consumption. If this information is included in the model, if will probably improve the relevance of findings.
Author Response

(The authors gave the same response as above.)

Reviewer 3 Report
Dear authors,
It was a pleasure to review your manuscript. It certainly touches upon an interesting and growing topic. However, I have my reservations in terms of the quality of your manuscript. I will try to comprehensively present them in the next few lines:
1. Please be specific when it comes to the research gap. From your introduction, it is not rather clear what the gap is and why it is important to go after it. Additionally, please try to be more concrete when explaining why the study in your region should be performed. Just stating that so far it has not been the subject of investigation is not enough. You have to present clearly the rationale behind it.
2. Although I am not a native English speaker, I can see an abundance of grammatical, writing, and syntax errors all around the manuscript. Please forward your manuscript for proofreading once you are done with the revision.
3. I would advise restructuring the literature review. In its current shape, it is a bit confusing. Please make the literature review and individual section with individual sub-sections. Please refer to other contributions to reflect.
4. The moderating role of demographic factors is very shallowly presented. At some points, the way you hypothesize some relationships seems trivial. Moreover, I would appreciate it if you provide a rationale for each proposed hypothesis (H4a, H4b, etc.).
5. In terms of methodology, I am not very confident in the results that are based on convenience sampling. Although you have a solid number of respondents, this can be a marginal factor if the sampling is done wrong. Of course, you can not do anything at this point but mention this as a huge limitation of your study and call for replication using more robust data collection procedures.
6. In terms of implications, you are very vague. In a sense, you just mention or write the supposed contribution, without real arguments or reflecting on the literature if that is really so.
7. For the sake of transparency, reviewers would like to have attached the Spanish version of the survey.
In closing, I must express my optimism about this manuscript. But some fine-tuning is needed in order to have it publishable.
All the best.
Author Response

(The authors gave the same response as above.)

Reviewer 4 Report
The topic of the paper is very interesting, bearing in mind that it covers sustainable consumer intention and its predictors; which may have implications on both social and environmental aspects.
However, there are some issues that need to be resolved or explained in more detail.
When it comes to the introduction section, the subject of the research and its importance were explained. Hereby, the literature review section was included in the introduction as a subheading 1.1. The literature review section (and its subsections) should be set as a separate part 2.
When it comes to the research framework, the authors need to give a theoretical explanation or a proper theoretical background for choosing selected variables, i.e. to mention similar studies that combined those variables. For example, the TRA and the TPB theory, which were also mentioned in the research, include certain variables. Thus, in accordance with what theory or approach, the authors selected four analyzed variables?
In the case of variables, the dependent variable - Environmentally Responsible Purchase Intention (ERPI) needs to be defined. The authors explained the purchase intention, but they didn’t give an adequate explanation or a definition of what they assume under the ERPI.
Predictors of ERPI are very similar and in some cases may overlap; because of this, the authors should emphasize distinctions among them.
The problem may be hypothesis H2 or the relation between Sustainable Consumption (SC) and Environmentally Responsible Purchase Intention (ERPI). As stated in the TPB theory, the purchase intention predicts the behavior and not vice versa. Therefore, the authors should explain how the consumption can impact intention.
Hypothesis H3 misses the verb - Social Responsibility (SR) Environmentally Responsible Purchase Intention (ERPI).
Text in section 2. Materials and Methods (lines 233 - 240) should be deleted or changed as it refers to the guidelines for the authors.
The measurement items used in this research include the COVID-19 pandemic as the main „predictor“ of all other variables (Environmental Awareness, Sustainable Consumption, Social responsibility, and Environmentally Responsible Purchase Intention). Hereby, the relations between analyzed variables, for example between Environmental Awareness and ERPI, actually represent the relation between the impact of COVID-19 on Environmental Awareness and the impact of COVID-19 on ERPI. In addition, those measurement items are not in a line with the hypotheses in which COVID-19 isn’t even mentioned.
For testing the model and hypotheses, the authors relied on SEM. The results presented in Tables 3, 4, and 5 should be checked - not only the numbers but the labels as well. The explanation related to factor loadings (and the elimination of some items) misses. Mentioning the values of SD and Std. Beta, including ”***“ (presented in Table 3) should be explained. Table 4. contains CR and AVE values; and ”***“ next to some values. Also, both Tables 4. and 5. have labels AE instead of EA, and MR instead of SR.
The discussion and conclusion sections should be adjusted to potential preciously stated changes.
Author Response

(The authors gave the same response as above.)

Round 2
Reviewer 1 Report
The authors have made satisfactory improvements to the study by addressing the majority of the corrections outlined in the previous review. However, the impact of item exclusion on the results and conclusions of the study has not been fully elaborated. It is important to consider the theoretical implications of excluding items from constructs and the potential impact on the validity and reliability of the results.
Author Response

(The authors gave the same response as above.)

Reviewer 4 Report
Dear Authors,
The second version of your research is much better.
However, there are three issues that, in accordance to my opinion need to be explained:
1) the relation between sustainable consumption and intention should be explained better. You added two sentences, but in them the emphasis was on brand and label analysis during the shopping process, and not on the consumption. You can add a few studies in which the effect of consumption on intention was examined, including more adequate explanation.
2) concerning the COVID-19 context, you have changed the hypotheses. Still, most variables in the text were disscused without mentioning this aspect. In addition, you can change the name of the variables. For example, the items you used for measuring "sustainable consumption" are actually measuring sustainable consumption in the context of COVID-19.
3) concerning the methods and steps used in the analysis, you can check the researches of the authors Hair et al. related to SEM.
Best regards.
Author Response

(The authors gave the same response as above.)
